# Exceptional Illuminated Manuscripts at the Gulbenkian Museum: The Colors of a Bible and Three Gospels Produced in the Armenian Diaspora

Hermine Grigoryan [1,2], Márcia Vieira [1], Paula Nabais [1,2], Rita Araújo [1], Maria J. Melo [1,2,*], Marta Manso [3,4], Maria Adelaide Miranda [2] and Jorge Rodrigues [5]

1 LAQV-REQUIMTE and Department of Conservation and Restoration (DCR), NOVA School of Sciences and Technology, 2829-516 Monte da Caparica, Portugal
2 Institute of Medieval Studies (IEM), NOVA University of Lisbon, Av. Prof. Gama Pinto, 1646-003 Lisboa, Portugal
3 LIBPhys-UNL, Department of Physics, NOVA School of Science and Technology of NOVA University of Lisbon, 2829-516 Caparica, Portugal
4 VICARTE, DCR, NOVA School of Science and Technology, 2829-516 Caparica, Portugal
5 Calouste Gulbenkian Museum, Founder's Collection, Av. Berna, 45A, 1067-001 Lisboa, Portugal
* Correspondence: mjm@fct.unl.pt

**Abstract:** The illuminated manuscripts at the Gulbenkian Museum were produced in the 17th century, in scriptoria of the Armenian diaspora. In this work, we selected analytical methods that can be used in situ to study the colors of the illuminations. Scientific analysis based on fiber-optics reflectance spectroscopy in the visible and Raman spectroscopy has shown the use of a medieval palette based on inorganic pigments such as lapis lazuli, minium, vermilion, orpiment, indigo, two different greens (vergaut and malachite), lead white and carbon black. More importantly, in this context, it showed that the very important reds and pinks are possibly based on carminic acid. The painting technique is, however, different, as are the ways of painting the faces, hands, and vestments. The range of colors in the Bible and the three Gospel Books, enhanced by lapis lazuli blue and organic reds and pinks, demonstrates a desire to create exceptional illuminated manuscripts.

**Keywords:** Armenian manuscripts; Gulbenkian collection; dyes and pigments; spectroscopic analysis; painting techniques



## 1. Introduction

### 1.1. Illuminated Manuscripts Produced within the Armenian Diaspora in the 17th Century

Illuminated manuscripts comprise one of the major fields of Armenian art and receive scholarly attention steadily. The main hub for these precious artworks is Matenadaran Institute of Ancient Manuscripts in Yerevan. Apart from that, there are numerous Armenian manuscripts preserved in collections worldwide. To unveil these codices as hidden gems is such a delight for a researcher. Our group had the possibility to explore a collection of four Armenian manuscripts (LA152, LA193, LA216, LA253) preserved in the Gulbenkian Museum in Lisbon.

Four manuscripts were acquired by Calouste Gulbenkian during his lifetime from different art dealers [1]. They became a part of the permanent exhibition of the Gulbenkian Museum since the very beginning of its establishment. The collection includes one Bible (LA152) and three Gospel Books (LA193, LA216, LA253). The preserved colophons indicate the place and date for the manuscripts as follows: LA152, Constantinople, 1623; LA193, Crimea, 17th century; LA216, 1686, Isfahan (New Julfa). LA253 does not possess any colophon, but based on its miniature style it was attributed to the 17th century school of Constantinople. Each manuscript is unique on its own, with an interesting biography

and exquisite art of illuminations. Our larger discussion on manuscripts' art, history, and colophons can be found in [2,3].

As already mentioned, this group of manuscripts was produced within the Armenian diaspora communities of Constantinople, Isfahan, and Crimea, during the 17th century, as shown in Figure 1. These communities were prospering centers of Armenian trade and artistic production patronaged by the wealthy merchant class. The three centers were interconnected through the dynamic mobility of people that promoted the circulation of goods and ideas between them. At the same time, these communities shared wider connections globally, from East to West [4]. The artistic production of this period, developed beyond the historical frontiers of Armenia, is distinguished by its eclectic style. This style is strongly characterized by the continuity of Armenian medieval local traditions and contacts with European art conveyed by the Crusaders in the Latin Kingdoms of Cyprus, Antioch, and Jerusalem. It is in the Kingdom of Cilicia (1198–1375), open to the Mediterranean, that Armenian art has its apogee and where, in the 13th century, flourish the scriptorium of Hromkla (present-day Rumkale, Gaziantep, Turkey), and the legacy of Toros Roslin, scribe and illuminator of seven dated manuscripts. The inheritance of Byzantine and Seljuk art intersects here as well [5]. These broad connections can be particularly observed in the manuscript art of the 17th century Armenian diasporas, where the local artistic traditions are very often conjoined with new westernized inspirations [6,7]. It has been suggested that these diaspora workshops bear influences from traditional Armenian schools, particularly from Cilicia, as well as from Byzantine and Western art [8–10]. Therefore, the Gulbenkian group of Armenian manuscripts offers the possibility to study the manuscript production of the last Armenian scriptoria, and to evaluate the art and craftsmanship of miniature workshops of Constantinople, Isfahan, and Crimea.

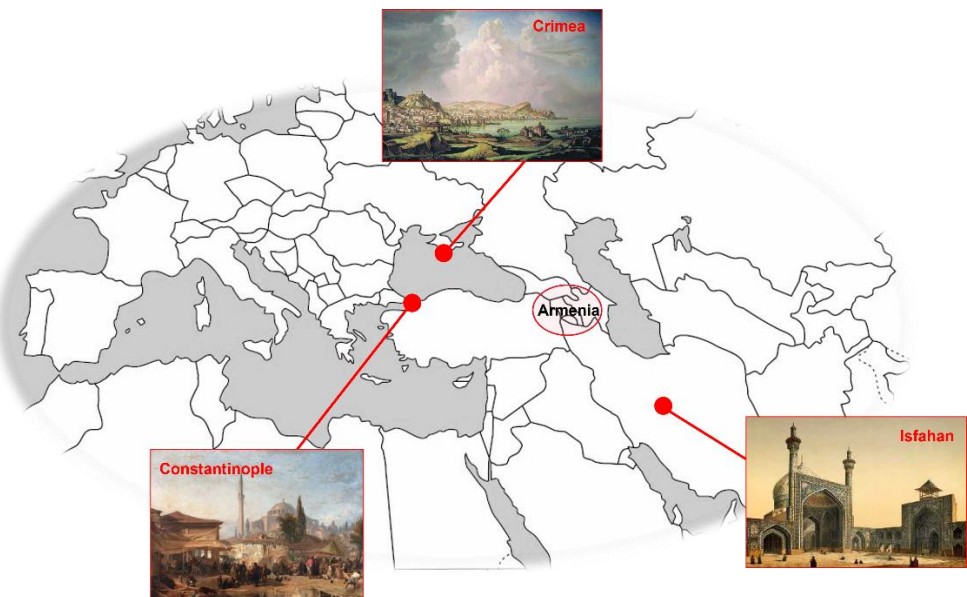

**Figure 1.** Localization of the manuscript production centers, Constantinople, Crimea, Isfahan (New Julfa), and present-day Armenia.

In the present study, we aim to explore the color palette applied in these four Armenian manuscripts and in future work to characterize the complete paint formulations, including binders and additives, Figures 2–5. Our specific interest is in the organic-based colorants, with hues ranging from pink to carmine and purple, which in future work we will study in depth to verify whether they were made with Armenian cochineal [11].

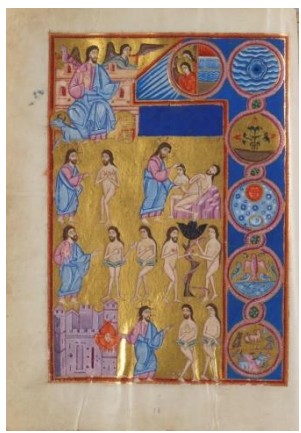
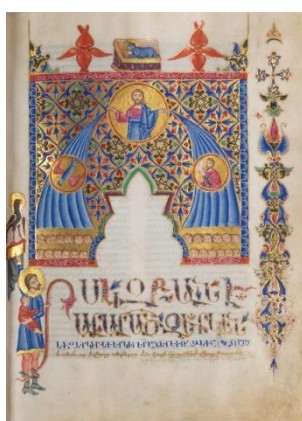

**Figure 2.** Bible LA152 (1623), produced in Constantinople (224 × 165 mm), from left to right: "Creation", p. 13; "Revelation" (title-page of the Book of Genesis), p. 14. © Gulbenkian Museum.

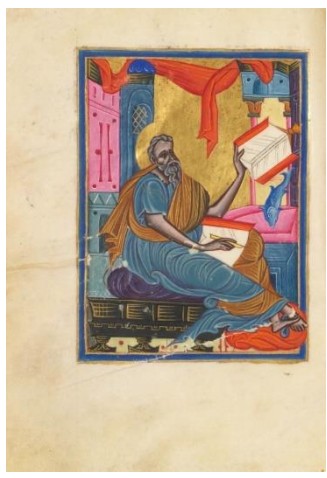
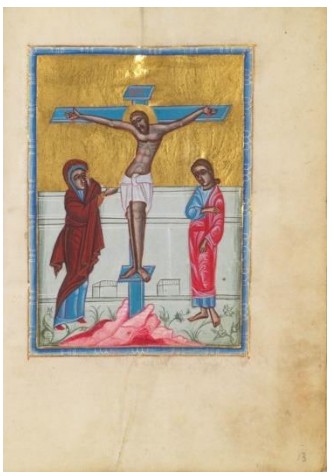

**Figure 3.** Gospel LA253 (17th century), produced probably in Constantinople (154 × 114 mm), from left to right: "Evangelist Mathew", f. 28v; "Crucifixion" f. 13r. © Gulbenkian Museum.

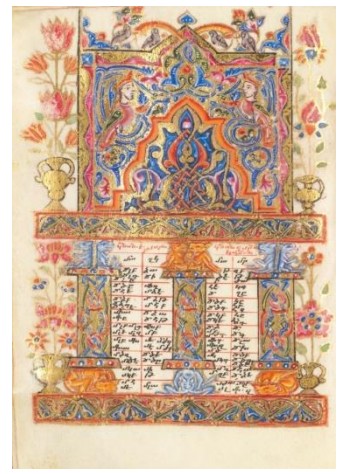
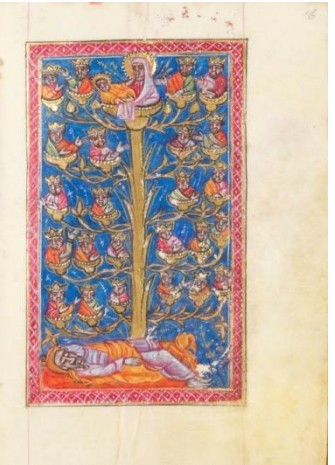

**Figure 4.** Gospel LA216 (1686), produced in Isfahan/New Julfa (108 × 079 mm), from left to right: "Canon Table" f. 5v; "Tree of Jesse" f. 16r. © Gulbenkian Museum.

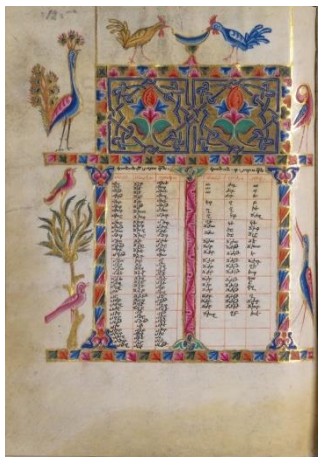 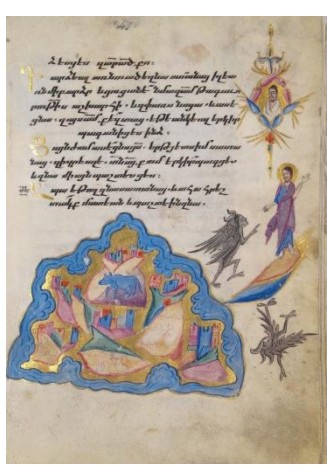

**Figure 5.** Gospel LA193 (17th century), produced in Crimea (176 × 133 mm), from left to right: "Canon Table" p. 12; "Temptation in the desert" p. 41. © Gulbenkian Museum.

## 1.2. Multi-Analytical Approach and Previous Studies

Identifying the colorants in the Armenian illuminations is based on the UV-Visible spectra acquired with FORS (Fiber-Optics Reflectance Spectroscopy) and Raman's spectra acquired with handheld equipment. The study starts with acquiring the UV-Visible spectra, which will guide the points of analysis by Raman. The data obtained will allow proposing a first molecular palette of the colorants used by comparison with paint reproductions prepared with as much historical accuracy as possible [12–15]. It is complemented with Energy dispersive X-ray fluorescence spectrometry (XRF) to confirm the presence of gold and other inorganic pigments. In future work, the complex formulations used in these colors will be explored using infrared spectroscopy and fluorescence spectra in the visible, which will make it possible to determine the binding media used and the specificities of the organic-based reds [12,15].

This work will compare the main colorants identified with previous studies on Armenian manuscripts summarized in Table 1. This table is based on the information collected in Tables S1 and S2. Previous studies exploring the materiality of Armenian manuscripts include important works by two main groups: Orna and co-workers, and Keheyan and co-workers, and complemented by other studies [16–30]. Orna, Cabelli and Mathews's collaboration were the first to study the pigments of Armenian illuminations systematically. Their examination of 24 manuscripts using mostly elemental analysis draws the overall image of the Armenian medieval palette, Table 1. This chronology expands by studies of Keheyan and co-workers, based on manuscripts from different miniature schools, and being the first to implement Raman microscopy in this process. Indeed, understanding the nature of all the components of such complex objects as codices, i.e., binding, parchment or paper, inks, and pigments, will widen our knowledge of the specificities of the production process, artistic techniques, paint and ink formulations, and much more. Such knowledge is indispensable for manuscripts' conservation and restoration and may contribute to a number of areas, ranging from art history and technology to socio-economic and cultural history.

**Table 1.** Selection of the main pigments described in Armenian manuscripts ranging in date from 10th to the 17th centuries, based on the works of Orna et al. in the 1980s, and Keheyan et al. in 2000s. Mostly inorganic pigments were identified, and, more infrequently, organic dyes.

| Most Used Pigments | Less Used Pigments | Mixtures |
|---|---|---|
| Lapis lazuli | Indigo | Green |
| Vermilion | Azurite | Yellowish green |
| Red lead (minium) | Smalt | Purple |
| Orpiment | Yellow ochre | Orange |
| Gold leaf | Organic yellow | Brown |
| Organic red | Malachite | Flesh |
| Lead white | Verdigris | |
| Carbon black | | |

*1.3. Overview of the Medieval European Palette*

Medieval colors used in illuminated manuscripts, from the 12–15th centuries, were based on a luxurious and restricted palette dominated by a certain number of inorganic pigments and organic dyes. The medieval codex, with the Bible as its foundation, started in the monastic scriptoria and evolved into lay workshops for the production of books of hours. During this period, some pigments were replaced, allowing us to discuss when these manuscripts were made within a specific context. In the monastic world, color also had a significant symbolic meaning [31–35].

In the most important manuscripts, lapis lazuli was the main blue [31,35,36]. Indigo was also used to shadow, as well as a single color or admixed with yellows or reds to create greens (vergaut) and purples [37,38]. Azurite was the other important blue [35].

For the reds, vermilion and red lead (minium) were the main inorganic pigments, and although minium is orange it was incorporated in the sphere of reds [31,35,39]. Lac dye and brazilwood produced pigment lakes ranging from dark reds, carmines, and pinks, which could be opaque or translucent [12,15,31,40,41].

Orpiment was one of the essential yellows in monastic production, being replaced by mosaic gold and lead-tin yellow from the 14th century onwards, being an important color in books of hours [31,35].

A durable green was a difficult color to produce in medieval times; in Bibles, Gospels, and other religious texts produced during the 8–12th centuries, there is evidence that a very saturated green color was used. It was identified as a proteinaceous copper green in Portuguese monastic production dated from the 12–13th centuries (it was named bottle green) [12,31]. Malachite and other basic copper sulfates were used in books of hours [12,35,42].

White was essentially produced with lead white, but there are examples of the application of calcinated gypsum (anhydrite) in precious manuscripts such as the Book of Kells [12,38]. Blacks were carbon black or bone black.

Gold and silver were also profusely used in important manuscripts, applied as leaves or inks.

Crucial for the preservation of these paints was their formulation, which includes the binding media, hereafter referred to as tempera. Tempera could be based on proteins or polysaccharides [12,14,43,44]. Medieval treatises also describe mixtures of both [14,43,44]. For medieval illuminations, protein-based tempera could be prepared using egg white (glair) or parchment glue; the use of small amounts of egg yolk is also sometimes described; it is possible that it was a relevant component in vermilion colors. The most important polysaccharide in medieval treatises is gum arabic [43,45]. We have evidence that other types of gums of this family, such as mesquite gum, were also employed [13,46,47]. Protective varnishes would have been applied and could have similar formulations to tempera.

## 2. Results and Discussion

### 2.1. The Molecular Palette Used to Produce the Colors of the Bible and the Gospels

The main colorants used in the four manuscripts studied are the same, and the molecular palette is represented in Figure 6. The painting technique is, however, different, as are different the ways of painting faces, hands, and vestments, which are discussed in Section 2.4. More details on the painting technique are found in Supplementary S1. The areas of analysis and spectral information are described in Supplementary S2 and S3, respectively.

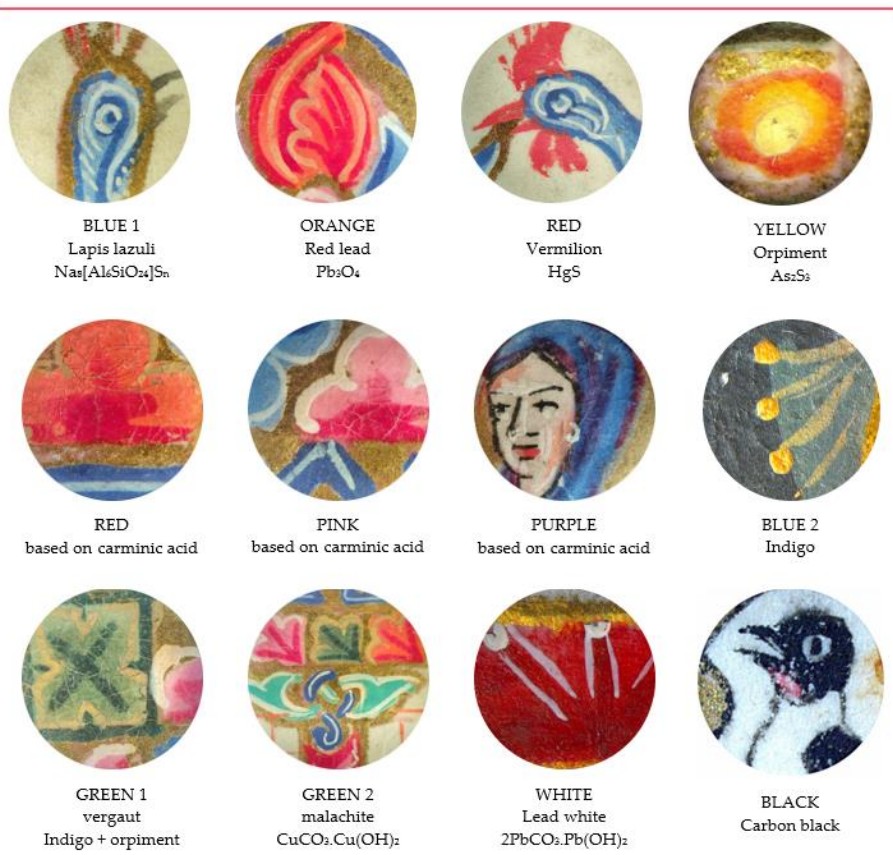

**Figure 6.** The molecular palette for the studied manuscripts with details from Gospel LA193 (17th century, Crimea), as well as one, the blackbird, from Gospel LA216 (1686, Isfahan). Lapis-lazuli and *vergaut* are the main blue and green colors in three of the manuscripts; in Gospel LA253 (17th century, Constantinople) azurite is the main blue and in Gospel LA193 malachite is the main green.

**Lapis lazuli** is the main blue, except for Gospel LA253, in which azurite is also an important pigment in the illuminations. This azurite blue is usually used to paint the initials. Indigo is also applied to darken colors and to create light blues, admixed with lead white. The most important color is, possibly, the organic-based red, also used to create pinks and purples, as shown in Figure 7. It is used in the contours, as a ground for gold, to draw details over the faces, vestments, and architecture details, and to create many other decorations in the margins, Supplementary S1.1–S.4. The reflectance spectra acquired in the visible point out to a color based on carminic acid obtained from a scale insect [48]. The other **inorganic reds** are vermilion and minium. However, Raman spectroscopy detected essentially vermilion in orange and red colors, and minium was seldom identified. XRF data shows, on the other hand, that minium is ubiquitous, as shown in Supplementary S3.

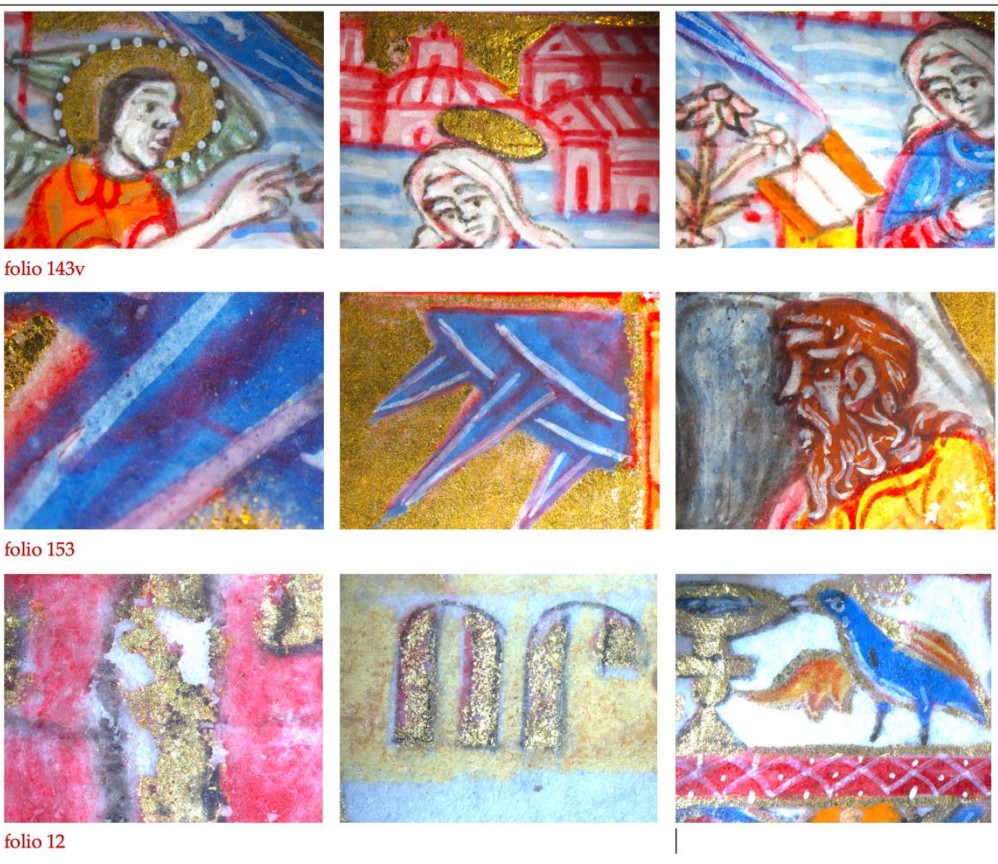

**Figure 7.** Application of organic-based reds/carmines/purples in Gospel LA216 (1686, Isfahan). Used as outline, ground layer and color (details obtained with a Leica microscope).

**Pinks** are prepared by admixing the organic-based red with lead white. In Gospel LA216 (Isfahan), calcium carbonate is also identified in addition to lead white to create the pink color. For purples, this red is combined with blue; it is lapis lazuli for the Bible and Gospels LA193 (Crimea) and LA216 (Isfahan), while in Gospel LA253 (Constantinople), it is indigo.

It is interesting to observe the use of the very poisonous **orpiment** as the main yellow. The conservation condition of this color in the Bible and the three Gospels is discussed in Supplementary S1.1.2, S.2.2, S.3.2, and S.4.2. Possibly, in some manuscripts, the rather pale yellow now observed was more saturated. The other yellow color identified is an ochre that is used in Gospel LA216 (Isfahan).

**Greens** and **blacks** will be only tentatively discussed in this paper, and will be further explored in future works. Both FORS and Raman spectroscopy detected two different greens, one based on vergaut and the other on malachite. Vergaut is obtained using orpiment and indigo, and it is the main green in the Bible LA152 and the Gospels LA216 and LA253 (produced in Isfahan and Constantinople, respectively). Malachite is the main green in Gospel LA193 (Crimea). Raman indicates that blacks are based on carbon blacks.

White is always **lead white**, and it is applied in a similar way to what was observed in medieval times; highlights created with it usually have a low amount of tempera, and inside, there is a void [12,49]. For this reason, it is easily detachable and, therefore, a fragile color. This white also creates lighter colors, such as pinks and light blues.

### 2.1.1. Gold

Gold is prominently used in the four manuscripts on backgrounds, ornaments, and text. Gold has a homogeneous appearance, especially in Bible LA152. However, there are

several areas of gold loss and detachment in both paint layers and written text. Pure gold (nor silver nor copper was detected) was identified using XRF.

### 2.1.2. Other Blues

Cobalt and other elements associated with cobalt minerals (nickel, arsenic, and bismuth) were identified in some of the blues present in the Gospel LA 216 (Isfahan). This blue was detected in the calf representing the evangelist Luke (141v) and in the vestment of the evangelist John (213v). Furthermore, potassium and silicon were also present, indicating the use of a blue cobalt-containing potassium glass pigment. Thus, a first hypothesis of the use of smalt could be anticipated. However, this needs to be confirmed with further research.

### 2.2. Fiber-Optics Reflectance Spectroscopy as First Screening for the Study of the Colorants

Fiber-optics reflectance spectroscopy allowed for the screening of the main colorants present. The blue colorants identified are lapis lazuli, indigo, azurite, and mixtures of these colorants (Figure S21, Supplementary S3). In LA152 and LA216, the only blue identified is lapis lazuli. LA193 also has blues prepared with a mixture of lapis lazuli and indigo. In LA253, azurite is also identified. In this manuscript, lapis lazuli is applied in the frames of illuminations and on some vestments; azurite is here the main blue and is used for vestments and architecture.

The greens are prepared using a mixture of a yellow and a blue colorant. Only in LA152 and LA193 is malachite identified as a source of a light green (Figure S22, Supplementary S3). In LA152, malachite is identified in floral details and frames, while for LA193, it is present in architecture and vestments. Vermilion is detected in the reds and oranges (Figure S23, Supplementary S3). It is identified in all manuscripts, either alone or in mixtures with a yellow pigment and also minium, as demonstrated by the shift of the inflection point to lower wavelengths (Figure S23, Supplementary S3).

The yellows of all manuscripts are prepared using a yellow pigment, possibly orpiment. A yellow ochre is only present in LA153, Figure 8a. In LA152 and LA216, an unidentified yellow color is also detected, which could be an organic yellow or an aged tempera or varnish, Figure 8b.

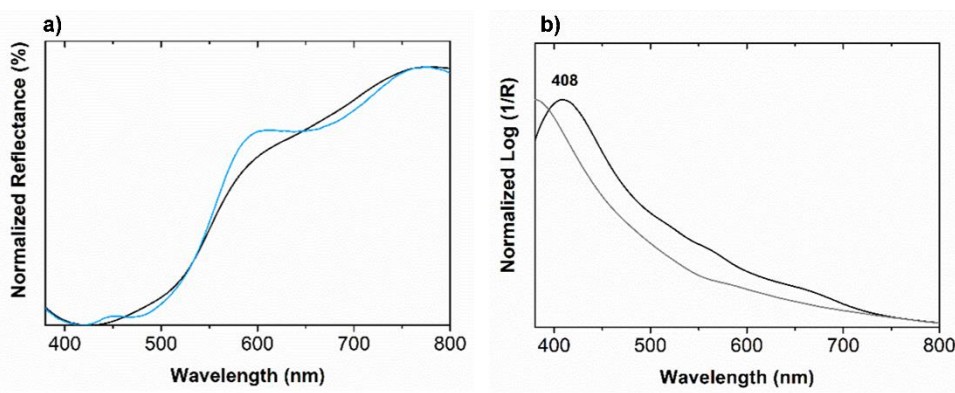

**Figure 8.** Apparent absorbance spectra of yellow colors, in (**a**) in LA253 f. 12v (**black line**), and a yellow ochre reference (blue line); (**b**) in LA216 f. 141v (**black line**) compared with parchment acquired in the same folio (grey line), to be identified.

The organic reds and pinks are identified as an anthraquinone-based dye from an animal source. FORS allowed for the distinguishing of two main colors: a pink and a red hue, both present in all manuscripts. The pink is characterized by an absorbance maximum at ca. 520–525 nm with a shoulder at 559–562 nm, and the red hue by an absorbance maximum at ca. 513–517 nm with a shoulder at 552–556 nm, see Figure 9a. The pink formulation is also used as a preparation layer for the gold (LA216) and as writing ink (LA193) (Figure 9b). Finally, the purples are identified as a mixture of an anthraquinone-based dye and two different blue colorants, possibly indigo and lapis lazuli, see Figure 9c,d.

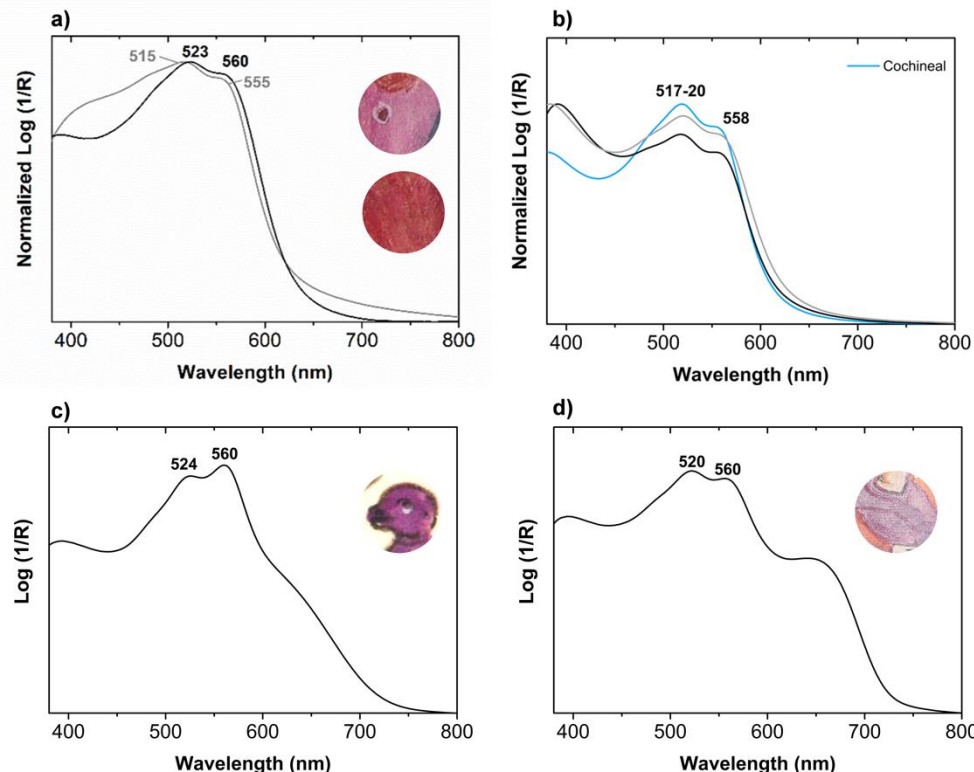

**Figure 9.** Apparent absorbance spectra of paints based on anthraquinone dye of animal origin: (**a**) pink color (**black line**) and red color (**grey line**), from LA193 p. 32; (**b**) applied as a preparation ground for gold, in LA216 f. 4r (**black line**) and as writing ink, in LA193 p. 168 (**grey line**), compared with a paint based on American cochineal (blue line); anthraquinone-based paints admixed with a blue colorant: (**c**) LA216 f. 4r and (**d**) LA193 p. 168.

The areas of analysis where the best signals were obtained with FORS were selected and analyzed with Raman spectroscopy to confirm and complete the information obtained.

### 2.3. Raman Spectroscopy for Identification of the Colors of the Bible and the Gospels

2.3.1. Inorganic Compounds

Raman spectroscopy allowed us to confirm some identification previously proposed by FORS, but also to identify mixtures of colorants and the use of lead white or fillers, such as calcium carbonate. It confirms the presence of lapis lazuli through lazurite in all four manuscripts and indigo in LA193 and LA253. In LA152, lapis lazuli is highlighted or lightened using lead white (Figure 10a). This is also the case with LA216, where calcium carbonate is also added to some blues. Two other mixtures found are lapis lazuli, indigo, and lead white, and another with the addition of calcium carbonate to the previous assembly (Figure 10b,c). See Figure 10d for examples of the application of these mixtures.

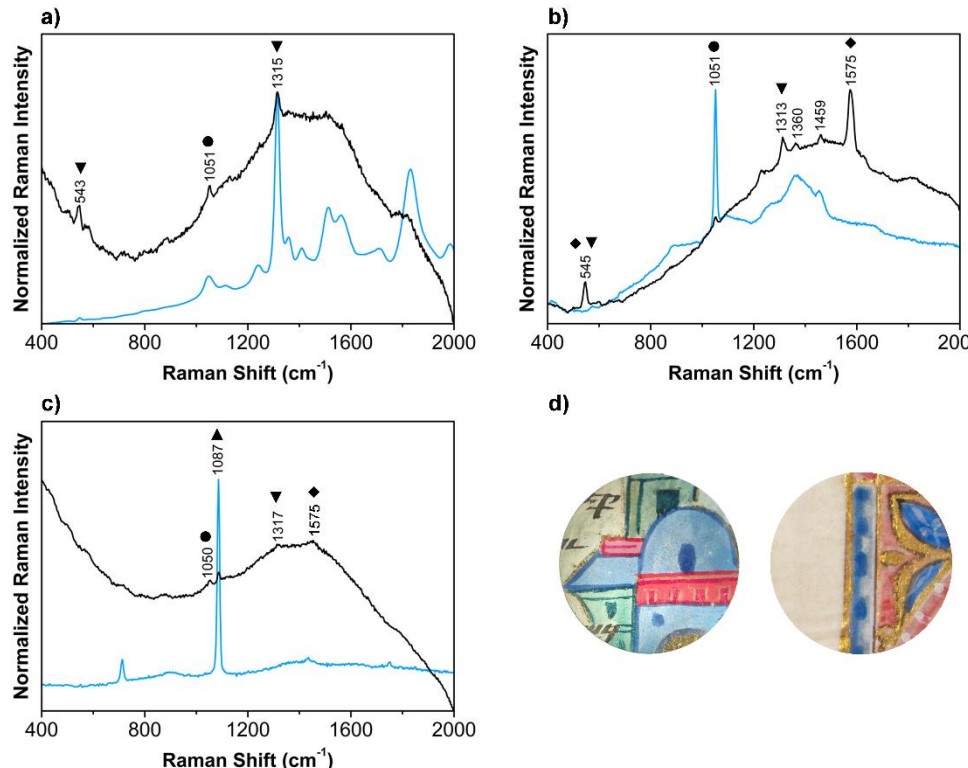

**Figure 10.** Raman spectra of the blue colors (black lines) are compared with references (ref.) (blue lines), prepared as gum arabic tempera and applied in parchment, for: (**a**) LA152, a mixture of lead white and lapis lazuli compared with lapis lazuli ref.; (**b**) LA193 and LA253, a mixture of lead white, lapis lazuli, and indigo compared with lead white ref.; (**c**) LA216, admixed with calcium carbonate lead white compared with calcium carbonate ref.; (**d**) details of the illuminations in LA193 (left) and LA216 (right). The main bands detected are assigned to lapis lazuli (▼), indigo (◆), lead white (●), and calcium carbonate (▲).

For the greens, three mixtures are identified: (i) orpiment and indigo, the most used in all four manuscripts, Figure 11a; (ii) orpiment and lapis lazuli, only present in LA152, Figure S24, Supplementary S3; and (iii) unidentified yellow and indigo, present in LA152 and LA193. Moreover, these colors were also prepared using lead white and/or calcium carbonate, as seen in LA193, Figure 11b.

For reds and oranges, vermilion is the main colorant used, as previously identified by FORS. Raman spectroscopy identifies the use of lead white to lighten the color in LA152 and LA193. Contrary to what is found in Portuguese medieval illuminations, minium is not found as a pure color, but always in a mixture with vermilion or orpiment, this last is represented in Figure 12a. Mixtures using vermilion are also identified: with orpiment, in LA152 and LA216 (Figure 12c); with minium, in LA152, LA193 and LA216 (Figure 12b). Moreover, the use of realgar is detected pure, in two manuscripts, LA152 and LA193, and in a mixture with orpiment, in LA216. The yellows, apart from those identified by FORS, orpiment, and a mixture of orpiment and realgar, are also identified (Figure 13a). Again, lead white is used to lighten the yellow from the orpiment, as shown in Figure 13b.

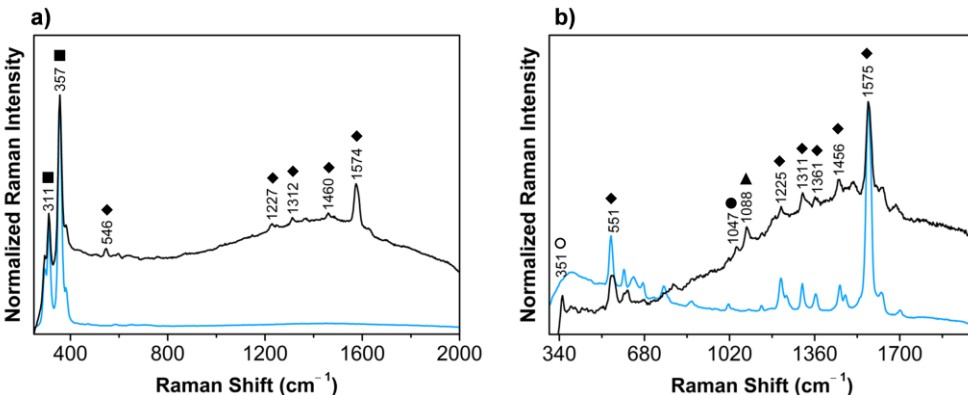

**Figure 11.** Raman spectra of two green colors (black lines) are compared with references (ref.) (blue lines), prepared as gum arabic tempera and applied in parchment, for: (**a**) *vergaut*, indigo and orpiment, which is identified in all manuscripts, compared with orpiment ref.; (**b**) LA193, as a mixture of realgar, calcium carbonate, lead white and indigo compared with the indigo ref. The main bands detected are assigned to indigo (♦), realgar (○), orpiment (■), calcium carbonate (▲) and lead white (●).

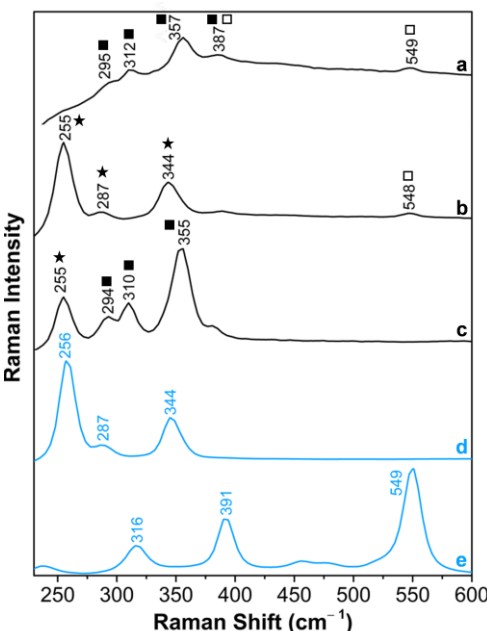

**Figure 12.** The red and orange pigments are identified as vermilion, the most used, and mixtures. The Raman spectra of three orange colors (black lines) found in: (**a**) f. 11v of LA216, as a mixture of orpiment and minium; (**b**) LA152, LA193, and LA216, as a mixture of vermilion and minium; (**c**) in LA152 and LA216, as orpiment with vermilion. The main bands detected are assigned to vermilion (★), orpiment (■) and minium (□). References (blue lines), prepared as gum arabic tempera and applied in parchment, for (**d**) vermilion and (**e**) minium.

The browns and greys are characterized by the presence of carbon black, in all manuscripts. In LA152 and LA193, the browns are prepared using minium (see Figure S25, Supplementary S3) and orpiment, respectively (Figure 14a). However, the addition of vermilion is more frequently detected (Figure 14b). For the grey colors, characterized only in LA216 and LA253, lead white (Figure 14d) and calcium carbonate (Figure 14f) (only in LA216) are added to the carbon black (Figure 14c,d).

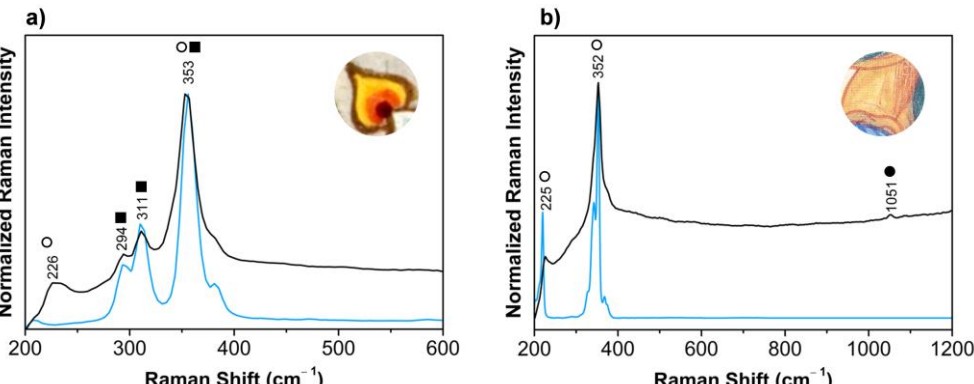

**Figure 13.** Orpiment and mixtures of this colorant with realgar and lead white were identified for the yellows. Raman spectra for the yellow colors (black lines) are compared with references (blue lines) for: (**a**) LA252 and LA216 as a mixture of realgar and orpiment; (**b**) LA193 as a mixture of realgar and lead white. The main bands detected are assigned to realgar (○), lead white (●), and orpiment (■).

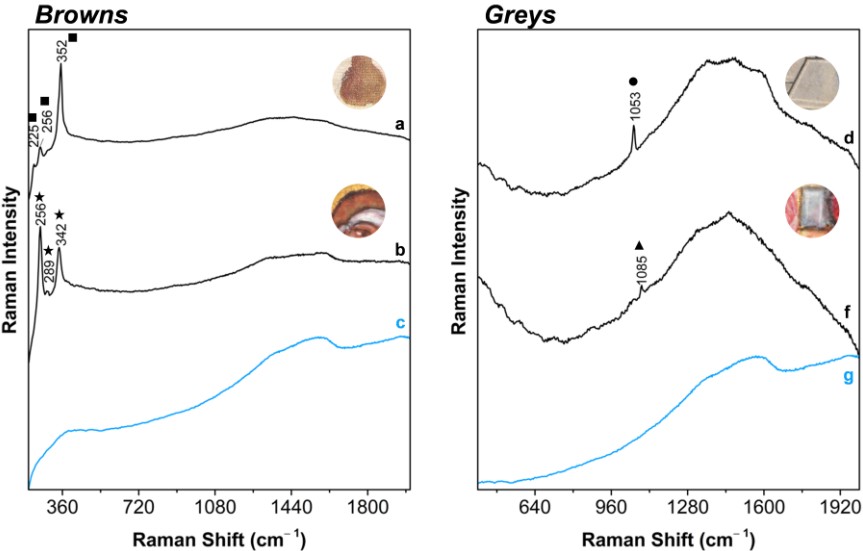

**Figure 14.** Raman spectra of the browns (**left**) and greys (**right**). The browns resulting from the mixture of carbon black with: (**a**) orpiment were found in LA193 and (**b**) vermilion was found in all the manuscripts. The greys were obtained using carbon black admixed with a white compound: (**d**) lead white in LA253 and LA216; (**f**) calcium carbonate for LA216. The main bands detected are assigned to orpiment (■), vermilion (★), lead white (●) and calcium carbonate (▲). Carbon black reference (blue lines), prepared as gum arabic tempera and applied in parchment, (**c**,**g**).

### 2.3.2. Reds and Pinks Based on Organic Dyes

The organic red and pink colors, previously characterized by FORS, are prepared with lead white and/or calcium carbonate, used as fillers or to lighten the color. Lead white is detected in LA152 and calcium carbonate in a mixture with lead white (Figure 15a) or alone (Figure 15b) in LA216. In LA152, a salmon color is characterized by a mixture of an anthraquinone-base dye (detected by FORS) and orpiment, Figure S26, Supplementary S3. The blue colorants used for the purple colors are lapis lazuli, in LA152 and LA216, and indigo, in LA193 and LA253 (Figure 15c,e). To this red and blue mixture, lead white is always added (reference in Figure 15f).

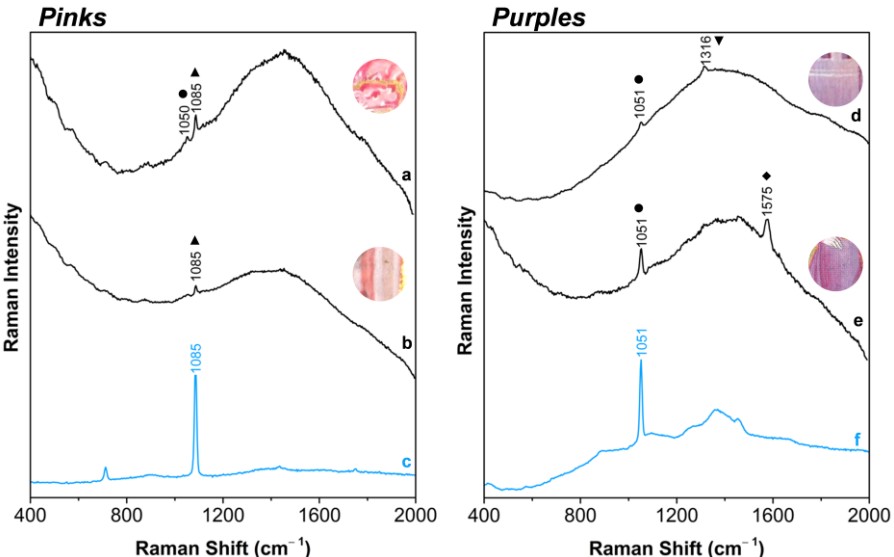

**Figure 15.** Raman spectra of the pinks and purples. The organic pinks were applied with (**a**) lead white and calcium carbonate in LA126 or with (**b**) calcium carbonate alone. The purples result from the mixture of a red dye and: (**d**) lapis lazuli in LA152 and LA216, (**e**) indigo in LA193 and LA253; and, except for LA216, all the paints present as additive lead white. The main bands detected are assigned to lead white (●), calcium carbonate (▲), lapis lazuli (▼), and indigo (♦). (**c**) Calcium carbonate and (**f**) lead white references (blue lines) are prepared as gum arabic tempera and applied in parchment.

Raman analysis of fluorescent dyes is complicated and is best done by SERS, which requires microsampling [50]. Therefore, with Raman spectroscopy, it was not possible to identify the organic red.

### 2.3.3. Writing Inks

The black writing inks are characterized as iron-gall inks through their Raman spectra. Iron-gall inks are prepared using a polyphenolic extract from galls, an iron salt, usually iron sulfate, and gum arabic to keep the precipitate formed in suspension. This formulation provides a very dark ink which has been used since antiquity. The main peaks that allowed for the identification of iron-gall inks are at around 500–600 cm$^{-1}$, 1335 cm$^{-1}$ and 1479 cm$^{-1}$, Figure 16. The blue and red initials were prepared using lapis lazuli and vermilion, respectively.

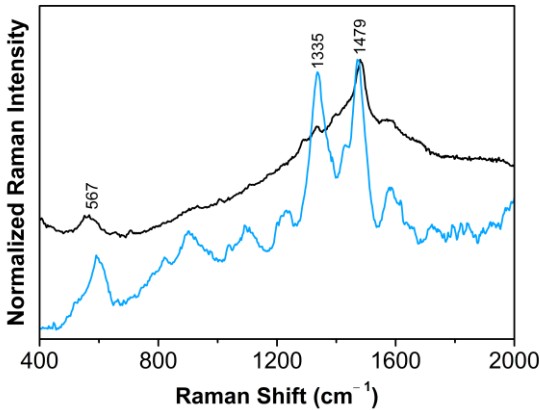

**Figure 16.** Raman spectrum of iron-gall ink found in f. 1210r, from LA152 (black line) and reference spectra of a reference iron-gall ink (blue line). This Raman spectrum is representative of iron-gall ink signals of all four manuscripts.

*2.4. Painting Technique*

The paratext in the four manuscripts is organized according to the text-image relationship commonly found in Armenian manuscripts—the main chapters and essential passages of sacred texts are usually indicated with marginal ornaments of vegetal patterns; the text begins with zoomorphic, anthropomorphic, ornated or bird-letters, Figure 17 and Supplementary S1 (Supplementary S1.5); titles and the following two-three lines are usually majuscule letters written in alternating red, blue, and gold inks; the main text is in black, with neat lines and regular calligraphy. In the case of the Bible, the narrow headpieces of interlaced vegetal patterns are provided at the beginning of the chapters, Figure 17. Content-related small narrative miniatures can be found within the text. The fully illuminated pages usually include the portraits and title-pages of the four Evangelists, the cycle of Eusebian letter followed by the Canon Tables, and the cycle representing the life of Christ typical for Gospel Books. Both figurative and ornamental illuminations are performed by skillful artists, each unique in its own way. As far as painting techniques are concerned, the artists seek to be as close as possible to their medieval counterparts, in how the colors are applied, dark and light shades are worked, or forms are modeled in these four manuscripts. The impression is that the ornaments, in general, were first outlined with red and then re-drawn with gold ink, as shown in Figure 7 and Supplementary S1 (Supplementary S1.3.1 and S1.5). A similar process was followed for the majuscule letters in gold, and for anthropomorphic, zoomorphic, or bird-letters. A similar approach was implemented in the outlines for the figures in LA193, where the gold was replaced by fine black lines so that both red and black outlines can be seen complementing one another, as shown in Supplementary S1 (Supplementary S1.4.1 and S1.5). The small marginal figures in LA253 seem to be re-drawn with black. This approach of the initial sketching of the illuminations with red is quite common in Armenian manuscripts.

A preserved Armenian painters' manual from 1618 (BnF MS Arménien 186) describes some drawing techniques that are traceable in our manuscripts [51,52]. These include mixing ochre with black and drawing the faces, hands, and feet, then re-drawing them with purple, or drawing with black and then shading it with red. It also suggests lightening the faces with a mixture of purple and white, or modeling the facial features by applying white and red on the noses, cheeks, foreheads, mouths, necks, hands, and feet. The manual mentions paint mixtures as well: red with black for deep purple, red with white for light red, red with lapis lazuli for light purple, red with indigo for medium purple, indigo with yellow for dark green, and lapis lazuli with yellow for light green. Finally, to create shades, it suggests indigo for lapis lazuli, light green for dark, and carmine for red.

Drawing lines and brushstrokes are delicate in LA152, LA253, and LA193, while they are thick in LA216, possibly because of the small dimensions of the book. A similar approach was also observed in the minute illuminations of a book of hours [53], as shown in Supplementary S1 (Supplementary S1.6). White is used to highlight the ornaments, vestments, mountains, and architectural details in all four manuscripts (Figures 17–19). More information on painting techniques is available in Supplementary S1.

Different technical approaches can be noticed in the full-page illuminations, particularly in portraying the figures, as shown in Figure 19. In LA152, the faces, bodies, and hair are drawn with brown. Hairs and beards are painted dark to light brown and greyish. Flesh tones are likely constructed on a light pink ground layer and modeled with abundant white and some reddish hues. Voluminous white spots and lines are prominent under magnification in highlighting noses, cheeks, foreheads, and the whites of the eyes. Mouths are marked with an orange/red single brushstroke. Hands and feet are highlighted with white and sometimes with pink, as shown in Figure 19 and Supplementary S1 (Supplementary S1.1.1).

In LA216, the faces, bodies, and hair are drawn in black or dark brown, as shown in Figure 19 and Supplementary S1 (Supplementary S1.2.1). Hair and beards are painted brown. Flesh tones are likely constructed on a dark brown ground layer and modeled with heavy white brushstrokes and black lines. Both dark brown and white layers are thick

and coarse in texture. Mouths are marked with double white lines. Hands and feet are highlighted with white as well. Unlike the other manuscripts, pink or red tones are not used for highlighting.

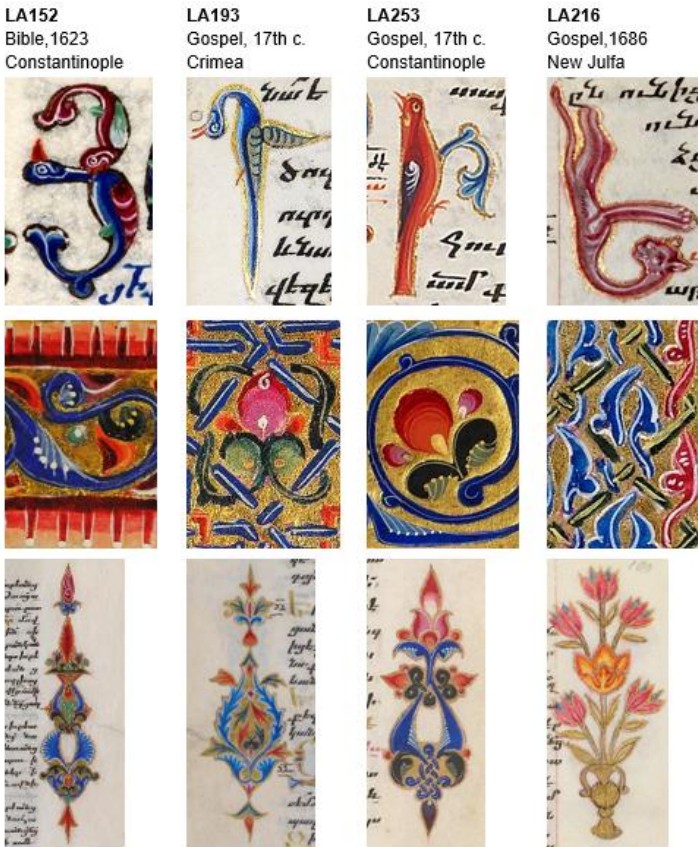

**Figure 17.** Details of incipit letters, headpiece patterns and marginal ornaments in the four manuscripts.

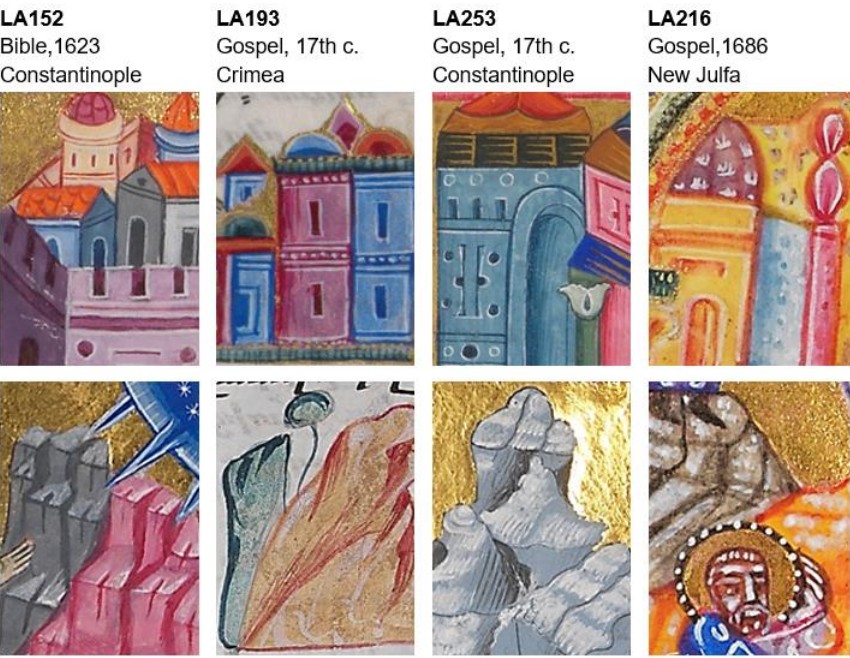

**Figure 18.** Details of architecture and mountains in the four manuscripts.

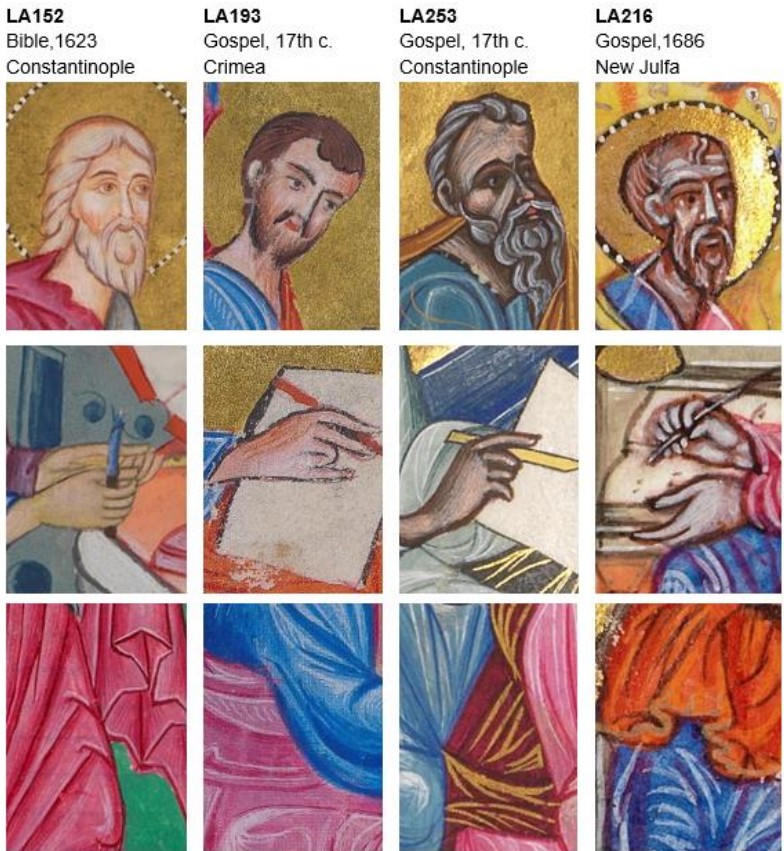

**Figure 19.** Details of faces, hands and vestments in the four manuscripts.

In LA253, the faces, bodies, and hair are drawn with thick brown lines, as shown in Figure 19 and Supplementary S1 (Supplementary S1.2.1). The eyes and eyebrows are marked with black. Hairs and beards are painted brown. Flesh tones are likely constructed on a dark brown ground layer and modeled with greyish white. Tiny pink brushstrokes are visible on cheeks, foreheads, noses, and mouths. Hands and feet are highlighted with greyish white and some pink. Under the microscope, tiny white dots are visible in the joints of the fingers, which is an exciting approach to creating the impression of flexibility. In terms of flesh tones, this manuscript resembles LA216 but with a less coarse texture and with the presence of pinks.

In LA193, the faces, bodies, and hair are drawn with smooth black lines and delicate forms, as shown in Figure 19 and Supplementary S1 (Supplementary S1.4.1 and S1.5). Sometimes, the pale red drawing lines are visible beneath the black lines. Hairs and beards are painted brown or grey. Flesh tones are likely constructed on a white or pale pink ground layer and modeled with red and white brushstrokes. Cheeks, foreheads, and necks are highlighted with red, animating the figures. In a few instances, mouths are marked with a single red brushstroke. Hands and feet are highlighted with some red. In terms of flesh tones, this manuscript resembles LA152.

To generalize, the flesh tones are light in Bible LA152 and Gospel LA193, primarily based on pinkish hues, while they are dark in Gospels LA253 and LA216, primarily based on brownish hues. Some creative details implemented by the artists give extraordinary expressivity to the figures. Drawings in the Bible LA152 are relatively static and well-defined, with clear lines and very smooth color transitions on painted areas, so the brushstrokes are almost untraceable. Gospel LA193 seeks to follow medieval mastery in the execution of the forms and color choices, such as with MS 7651 (Cilicia, 13–14th centuries) from the Matenadaran collection. Here, refined brushstrokes render the color manipulations and slender gestures, particularly of the hands. Gospel LA253 is an illustrious claim to classical forms closely interrelated with Byzantine art, particularly in portraying the Evangelists,

and is the unequivocal result of a medieval prototype such as MS 2629 (Cilicia, 13th century) from the Matenadaran collection. Gospel LA216, the small book, represents a unique artistic hand that gives an impression of spontaneous yet very ingenious solutions. The voluminous brushstrokes in modeling the figures and faces create a powerful visual effect. The artist seeks novel inspirations that probably originate from western schools. Although there are some similarities in the rendering of the ornaments, these manuscripts have entirely different methods to figure constructions and flesh tones that will be interesting to explore further. The diversity of Armenian miniature schools could contribute to the dating of manuscripts.

Among the four manuscripts, we highlight the elegance of the painting on the Bible LA152, which reflects the vision of its master, who knew how to combine techniques from the West and the East and could count on a prosperous commissioner interested in praising Armenian art and culture by recreating a golden era through illuminated manuscripts.

## 3. Materials and Methods

### 3.1. Points of Analysis for the Four Manuscripts

The areas of analysis by Portable Raman, FORS, and XRF are described in Supplementary S2. Our methodology involves analyzing at least three points by XRF and FORS per color hue (e.g., light blue, dark blue). This analysis will allow us to select the best areas to analyze via Raman, with at least one point per color. The folia were selected according to their interest and to have the widest range of colors possible, representative of the manuscript. This was the case for the analysis done in LA152 (pp. 13, 14, 588, 631, 795, and 796), LA193 (pp. 32, 160, and 168), LA216 (ff. 4r, 11v, and 141v), and LA253 (ff. 5v, 9r, 12v, 19v, and 28v). After this screening, it was necessary to understand if the characterization of some colors was consistent throughout the manuscript (e.g., the anthraquinone pinks and reds). Therefore, more punctual analyses were done using the technique best suited, in LA152 (pp. 15, 509, and 1210), LA193 (pp. 5, 12, 33, 97, 98, 134, and 171), LA216 (ff. 22v, 23v, 145r, 213v, and 217r), and LA253 (ff. 91v, and 277v).

### 3.2. Preparation of Historic Paint and Ink Reconstructions

The paint references used to compare and identify the colorants were produced with the following materials: Vermilion (May & Baker LTD Dageham England), malachite and orpiment (Kremer), Azurite (Zecchi), and reagent grade chemicals such as indigo and calcium carbonate (Aldrich), minium (Panreac). Lead white was prepared in our laboratory following the medieval process. Gallnuts (from *Quercus infectoria*) and gum arabic in grains from *A. senegal* were purchased from Kremer, and the last was prepared in a 20% solution in Millipore water.

The paint references were then made using 84% of gum arabic to 16% of pigment in weight and applied with an nº2 brush tool in circles of 2 cm$^2$ on parchment.

The iron-gall ink reference was made following a 15th century recipe found in the Archivo Histórico Provincial de Córdoba, Sección de Protocolos Notariales de Córdoba. The recipe is described elsewhere [54]. The paint was applied with a micropipette (60 μL) in squares of 2 cm$^2$ on filter paper.

### 3.3. Fiber-Optics Reflectance Spectroscopy

UV-VIS reflectance spectra were obtained with an Ocean Optics, MAYA 2000 Pro reflectance spectrophotometer equipped with single beam optical fibers and a Hamamatsu linear silicon CCD detector collecting spectra in a 200–1060 nm spectral range. The light source was an Ocean Optics HL-2000-HP halogen lamp, with 20-W output and 360–2400 nm spectral range. Analysis was conducted with a 8-ms integration time, 15 scans, eight box width, and 45°/45° reflection angle to the bearing surface, with a 2-mm spatial resolution. A Spectralon® white reference was used for calibration. While being acquired from 350 to 800 nm in reflectance, the spectra are shown as apparent absorbance, A' = log10 (1/R).

### 3.4. Portable Raman Spectroscopy

Handheld Raman spectroscopy was carried out with a Raman Mira DS, equipped with a laser emitting light at 785 nm with a maximum power of 100 mW, within a spectral range of 200–2300 nm. This equipment provides a spectral resolution of 8–10 cm$^{-1}$ and features a measuring spot of 0.042–2.5 mm. The detection technique used goes under the name of Orbital Raster Scan (ORS) and involves averaging the signal collected from relatively large sample areas while maintaining the desired resolution. All spectra were acquired with the maximum laser power and averages, varying the integration time according to the target material and working distance (the higher the distance, the higher the acquisition time). A minimum of three spectra were collected from the same sample to ensure reproducibility.

### 3.5. Portable Energy Dispersive X-ray Spectrometry

An energy dispersive X-ray spectrometry (XRF) analysis of the manuscripts was undertaken using a portable setup equipped with the Mini-X X-ray generator and the 123 SDD detector from Amptek®. The X-ray generator has an Rh anode. The outgoing radiation is collimated by a 1 mm diameter hole brass collimator with an aluminium insert. The silicon drift detector (SDD) has a 25 mm² detection area, a 500 mm thickness, and a 12.5 μm Be window. The detector energy resolution is 130 keV at 5.9 keV. The angle between the incident and the emitted X-ray beam is 90°, allowing for a high background reduction due to Compton scattering. The 4 mm distance between the spectrometer and the manuscript was controlled by a millimetre screw.

The manuscripts were analysed on air at room temperature. Spectra were acquired at 40 kV and 30 μA for 120 s using DppPMCA digital acquisition software from Amptek®. The deconvolution of X-ray spectra for peak location and peak area determination was performed using WinAxil (Analysis of X-rays by Iterative Least squares) from Canberra®.

## 4. Conclusions

In this work, we explored the materiality of a group of illuminated manuscripts produced in the last scriptoria of the Armenian diaspora, housed in the Gulbenkian Museum, and shown in Figures 2–5. It revealed a rich color palette behind the exquisite art of their unique illuminations. FORS and Raman spectroscopies allowed for the identification of the colorants present in the paints. We stress the use of lapis lazuli and the reds-pinks-purples based on carminic acid obtained from a scale insect, together with orpiment, as shown in Figure 6. The writing inks are based on iron gall inks. The color palette of pure and mixed pigments indicates the taste and preference of the 17th century Armenian merchant communities and the availability of certain materials likely circulating between Constantinople, Isfahan/New Julfa, and Crimea, as shown in Figure 1. At the same time, this palette indicates the continuity of medieval traditions in the early modern scriptorial practices through the use of medieval colorants in the illuminations such as lapis lazuli, vermilion, minium, orpiment, organic reds, mixtures of blue with yellows, and organic reds to create greens and purples, lead white, carbon black, and gold.

The illuminations and the painting technique are also discussed. Despite sharing common pigments, however, the mode of application is different for each of the four manuscripts, especially when comparing the paint layers and drawing outlines for figures and faces. Both the materials and the techniques implemented in these codices attest that the manuscript art of the Armenian diaspora remains deeply traditional, regardless of new inspirations. These manuscripts manifest the tribute of the 17th century Armenian scriptoria to the once glorious medieval art. At the same time, they represent quite individual approaches to artistic creativity.

As this article is devoted primarily to the color palette and painting techniques, we briefly discuss each manuscript's art, history, and codicology. These are based on previous [2,3] and current studies (the art historical study dedicated to three Armenian Gospel Books from the Gulbenkian collection is in preparation as a forthcoming publication).

This research will be continued in order to disclose the complete paint formulations and better understand the nature of anthraquinone-dye compositions. The use of red lead (minium) will also have to be studied further. Moreover, this study is essential in Armenian manuscripts and will pave the way for more comprehensive discussions.

**Supplementary Materials:** The following supporting information can be downloaded at: https://www.mdpi.com/article/10.3390/heritage6030170/s1, Figure S1: *Manuscript 23* (1410-1430), *f.* 16 (before intervention). Mafra National Palace collection.; Figure S2: *Manuscript 23* (1410-1430), *f.* 16. PNM collection. Details obtained with Leica microscope.; Figure S3: Bible LA 152, pages 13 and 14, © Gulbenkian Museum.; Figure S4: Bible LA 152, pages 15 and 509, © Gulbenkian Museum.; Figure S5: Bible LA 152, pages 588 and 631, © Gulbenkian Museum.; Figure S6: Bible LA 152, pages 795 and 796, © Gulbenkian Museum.; Figure S7: Bible LA 152, page 1210, © Gulbenkian Museum.; Figure S8: Gospel LA 193, pages 5 and 12, © Gulbenkian Museum.; Figure S9: Gospel LA 193, pages 32 and 33, © Gulbenkian Museum.; Figure S10: Gospel LA 193, pages 97 and 98, © Gulbenkian Museum.; Figure S11: Gospel LA 193, pages 134 and 160, © Gulbenkian Museum.; Figure S12: Gospel LA 193, pages 168 and 171, © Gulbenkian Museum.; Figure S13: Gospel LA 216, folia 4r and 11v, © Gulbenkian Museum.; Figure S14: Gospel LA 216, folia 22v and 23v, © Gulbenkian Museum.; Figure S15: Gospel LA 216, folia 141v and 145r, © Gulbenkian Museum.; Figure S16: Gospel LA 216, folia 213v and 217r, © Gulbenkian Museum.; Figure S17: Gospel LA 253, folia 5v and 9r, © Gulbenkian Museum.; Figure S18: Gospel LA 253, folia 12v and 19v, © Gulbenkian Museum.; Figure S19: Gospel LA 253, folia 28v and 91v, © Gulbenkian Museum.; Figure S20: Gospel LA 253, folio 277v, © Gulbenkian Museum.; Figure S21: Reflectance spectra of blue colors: *right*, lapis lazuli (black), LA152 p. 13, mixture with indigo (grey), LA253 f. 9r; *left*, azurite, LA253 f. 91v (black), compared with a reference of azurite applied on parchment (blue).; Figure S22: Reflectance spectrum of malachite, from LA152 p.14.; Figure S23: Reflectance spectra of orange-red colors: right, vermilion (black), LA253 f. 19v, and a mixture of vermilion and minium (grey), LA152 p. 14. The insert shows the first derivative which allows us to identify the inflection point. Left, mixture of minium and orpiment, LA216 f. 11v.; Figure S24: Raman spectra of green color produced with orpiment (■) and lapis lazuli (▼), from LA152 p. 13.; Figure S25: Raman spectra of brown color found in LA152 p. 795, composed by minium (□) and carbon black.; Figure S26: Raman spectra of salmon color found LA152 p. 13, composed by cochineal, previously identified by FORS, and orpiment (■).; Figure S27: XRF spectrum obtained at a gilded area from Bible LA 152.; Figure S28: XRF spectrum obtained at a blue area from Gospel LA 216.; Table S1: Pigments identified in Armenian manuscripts.; Table S2: Pigment mixtures identified in Armenian manuscripts.; Table S3: Net count areas and standard deviation for the oranges' key elements characteristic peaks.; Table S4: Net count areas and standard deviation for the yellows' key elements characteristic peaks.; Table S5: Net count areas and standard deviation for the greens' key elements characteristic.

**Author Contributions:** M.J.M. and M.A.M. conceived this research. M.A.M., M.J.M. and H.G. were in charge of the historical introduction and the visual characterization of the collection. M.J.M. and J.R. coordinated the access and analysis of the manuscripts. M.V. and P.N. were responsible for the studies using handheld Raman and fiber-optic reflectance spectroscopy. M.M. was responsible for the studies using the portable XRF. H.G. and R.A. participated in data acquisition and discussion. All authors have read and agreed to the published version of the manuscript.

**Funding:** This research was funded by the Portuguese Foundation for Science and Technology [Fundação para a Ciência e Tecnologia, Ministério da Educação e Ciência (FCT/MCTES)], through PhD grants awarded to Márcia Vieira [SFRH/BD/148729/2019] and to Hermine Grigoryan [PD/BD/142866/2018], CEEC junior contract awarded to Paula Nabais (2021.01344.CEECIND), UIDB/EAT/00729/2020, UIDP/00729/2020; Associate Laboratory for Green Chemistry- LAQV financed by FCT/MCTES (UID/QUI/50006/2019 and UIDB/50006/2020) and co-financed by the ERDF under the PT2020 Partnership Agreement (PO-CI-01-0145-FEDER-007265); Glass and Ceramic for the Arts—VICARTE financed by FCT/MCTES (UIDB/00729/2020 and UIDP/00729/2020); Laboratory for Instrumentation, Biomedical Engineering and Radiation Physics—LIBPhys financed by FCT/MCTES (UIDB/04559/2020 and UIDP/04559/2020). Calouste Gulbenkian Foundation grant on Armenian Studies (No. 269685) awarded to Hermine Grigoryan.

**Acknowledgments:** We are deeply grateful to the Director of the Calouste Gulbenkian Museum, António Filipe Pimentel, the Deputy Director, João Carvalho Dias, and all the collaborators who supported us during the on-site mission to study the four Armenian manuscripts.

**Conflicts of Interest:** The authors declare no conflict of interest. The funders had no role in the design of the study; in the collection, analyses, or interpretation of data; in the writing of the manuscript, or in the decision to publish the results.

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
