# Peer review of "Exceptional Illuminated Manuscripts at the Gulbenkian Museum: The Colors of a Bible and Three Gospels Produced in the Armenian Diaspora"

_heritage, doi:10.3390/heritage6030170_

Round 1
Reviewer 1 Report
A well-written and clear paper showing novelty and development of the techniques applied to study Armenian manuscripts from the Gulbenkian Museum for the first time. The obtained data will be useful for revealing history of paintings and beyond, and it may impact the corresponding field of study. The authors cite current relevant literature and discuss that literature in the context of their own results. I found Supplementary Material of this manuscript is particularly useful. I recommend that the manuscript be accepted with the minor revisions.
In this manuscript, the authors utilize FORS and Raman spectroscopies to identify pigments in four Armenian manuscripts. These two techniques are complementary and provide useful information. In particular, Raman spectroscopy is non-invasive and non-destructive approach, and its spectra are easy to interpret. The data obtained by FORS is less specific. It would be useful if the manuscript had a small section comparing two spectroscopic techniques by discussion about their advantages and limitations.
There are some minor things/questions that could be improved and/or answered regarding the manuscript.
Page 2 line 56, 57 add “as shown in” before Figure 1.
Page 5 line 131 This sentence is vague. Explain why it is important. One of the most common blue pigments?
Page 5 line 132 For the reds, vermilion and red lead (minium) were the main inorganic pigments,” it should be an independent sentence.
Page 5 lines 140-143 Too long sentence, it should be independent sentences.
Page 1-5 Section 1 The author may consider placing introduction/background and methods separately.
Page 6 line 175, 176 It needs to be rephrased. It looks like you are questioning readers.
Page 7 line 204-207 Check the font of this paragraph. The words look smaller than any other word in the texts. OK
Page 12 337-338 Superscript “cm-1”
Author Response
A well-written and clear paper showing novelty and development of the techniques applied to study Armenian manuscripts from the Gulbenkian Museum for the first time. The obtained data will be useful for revealing history of paintings and beyond, and it may impact the corresponding field of study. The authors cite current relevant literature and discuss that literature in the context of their own results. I found Supplementary Material of this manuscript is particularly useful. I recommend that the manuscript be accepted with the minor revisions.
In this manuscript, the authors utilize FORS and Raman spectroscopies to identify pigments in four Armenian manuscripts. These two techniques are complementary and provide useful information. In particular, Raman spectroscopy is non-invasive and non-destructive approach, and its spectra are easy to interpret. The data obtained by FORS is less specific. It would be useful if the manuscript had a small section comparing two spectroscopic techniques by discussion about their advantages and limitations.
There are some minor things/questions that could be improved and/or answered regarding the manuscript.
Answer: We thank the reviewer for constructive suggestions. Below is a point-by-point response; the text was revised accordingly. Moreover, we have rewritten certain sections, including the Painting technique and Conclusions, for clarity. The legends used in the spectra were also revised. The most relevant changes are marked in blue.
1) Page 2 line 56, 57 add “as shown in” before Figure 1.
Answer: It has been added.
2) Page 5 line 131 This sentence is vague. Explain why it is important. One of the most common blue pigments?
Answer: I am sorry, but only today I understood that we had a third reviewer, now Reviewer 1. So, I would prefer to leave this sentence as it is now.
3) Page 5 line 132 For the reds, vermilion and red lead (minium) were the main inorganic pigments,” it should be an independent sentence.
Answer: Please see my response to 2).
4) Page 5 lines 140-143 Too long sentence, it should be independent sentences.
Answer: They are now two independent sentences as follows:
A durable green was a difficult color to produce in medieval times; in Bibles, Gospels, and other religious texts produced during the 8th-12th centuries, there is evidence that a very saturated green color was used. It was identified as a proteinaceous copper green in Portuguese monastic production dated from the 12th-13th centuries (naming it bottle green) [11, 30].
5) Page 1-5 Section 1 The author may consider placing introduction/background and methods separately.
Answer: I am sorry, I could not understand.
6) Page 6 line 175, 176 It needs to be rephrased. It looks like you are questioning readers.
Answer: The sentence is now:
However, Raman spectroscopy detected essentially vermilion in orange and red colors, and minium was seldom identified. XRF data shows, on the other hand, that minium is ubiquitous, Appendix C.
7) Page 7 line 204-207 Check the font of this paragraph. The words look smaller than any other word in the texts. OK
Answer: Thank you, this has been corrected.

Reviewer 2 Report
February 12, 2023
The manuscript untitled “Exceptional illuminated manuscripts at the Gulbenkian Museum: a Bible and three Gospels produced in the Armenian diaspora” presents research on colors used in the illuminations of the manuscripts dated to the 17th century. By using fiber-optics reflectance spectroscopy and Raman spectroscopy Authors were able to identify pigments present in four objects produced by Armenian diaspora communities of Constantinople, Isfahan, and Crimea. I read the manuscript with an interest and I think it can be interesting also to those in non-destructive techniques in pigment analysis, human archaeology and conservation science. Nevertheless, I have comments and suggestions which I consider important and I kindly ask Authors to take it into account.
1. The title does not introduce the type of research presented in the paper.
2. Authors presents thesis that organic reds were made of Armenian cochineal, but they did not present any specific proof of that.
3. Section 2.1 – I suggest moving this section before the sections “Painting technique”, as well as changing the Appendix A into Appendix C.
4. Appendix B presents the areas of analysis by Portable Raman and by FORS. I suggest adding more detailed information about how many areas of each colour were examined on each page/object. Maybe Authors could put numbers or other code system of the areas in the presented pictures and give their detailed list below.
5. Appendix A – numbers of sections and subsections are confusing. I suggest numbering second-level headings as follows: 1.1, 1.2 etc., 2.1, 2.2 etc., 3.1, 3.2 etc., and 4.1, 4.2 etc.
6. Line 165 – I suggest removing the word "other" from this sentence.
7. Line 172 – Authors indicated Appendix A, but in my opinion, it needs to be more specific, for example “(Appendix A, section 3.1)”
8. Section 2.1 – whole section is a little difficult to follow. In my opinion, this section would benefit from presenting the similarities and differences in the pigments used in the manuscripts in a graphical form, e.g. in the form of a Venn diagram, radar plot or bar chart, using the actual number of areas studied or as a percentage (relative to the total number of areas in the object).
9. Lines 201-203 – the sentence “Lapis-lazuli and vergaut are the main blue and green colors in three of the manuscripts; in Gospel LA253 (17th century, Constantinople) azurite is the main blue and in Gospel LA193 malachite is the main green” in the caption of figure 6 is redundant. The same information is in section 2.1 (lines 189-192).
10. Line 204 – why does "Gold" have its own unnumbered subsection in section 2.1 when other pigments/dyes do not.
11. Lines 213-215, “Figure S1, Appendix C” – Authors refer to the same figure three times in only three lines of text. Is it necessary?
12. In addition, the figures in Appendix C are numbered C1, C2, C3, etc., not S1, S2, S3. This need to be corrected throughout the whole manuscript.
13. Line 218, “(…) see Appendix B and C” and line 222, “Appendix C” – Please, add a precise reference to the figures.
14. Line 222, “Appendix C” – Is this reference necessary?
15. Lines 227-230 and Figure 7 – I suggest adding points a) and b) in the figure (as in Figure 11) and refer to them in the text. This will improve the clarity of this fragment of text. This comment also applies to the other figures in the main body of the manuscript and in Appendixes B and C (instead of using the words "left" and "right").
16. Figure 7 – The caption is unclear and probably incorrect. What is the grey line on the left figure? The right figure shows only black and blue-lined reflectance spectra, no grey one.
17. Figure 8 – I suggest changing part of the caption into “(…) pink color (black) and red color (grey), both from LA193 page 32v (…)”.
18. Lines 236-239 – Were the obtained data compared with the references, as was the case in other cases? Why are the reference spectra not shown? Were they compared to the Armenian or Polish cochineal references, or even the American cochineal one? This is invariably important, especially in the context of the authors' thesis on the use of Armenian cochineal (see comment 2). Or maybe it would be worth discussing the differences between the spectra obtained for the studied areas and the references of lac dye, kermes or madder? Or maybe it would be worth discussing the differences between the spectra obtained for the studied areas and the references of lac dye or kermes? Moreover, this part of the paragraph suggests that pinks and reds give differences in absorbance maxima, but the left part of Figure 8 contradicts this. Perhaps these differences in the spectra result from the use of different anthraquinone-based dye of animal origin or the degree of their degradation. Therefore, I think there are clear gaps at this part of the revised manuscript that need to be completed.
19. Figures 8, 9 and C6 (Appendix C) – the captions mention “cochineal” but in the main body of the manuscript it is called “anthraquinone-based dye”. Since it has not been conclusively proven that this was a specific type of cochineal, I suggest sticking with the latter one.
20. Lines 242-244 – The sentence “The areas of analysis where the best signals were obtained with FORS were selected and analysed with Raman spectroscopy to confirm and complete the information obtained” should be a new paragraph.
21. Sections 2.2 and 2.3 – The headings could be changed and shortened. I suggest “Fiber-optics reflectance spectroscopy as a first screening for the study of pigments” and “Raman spectroscopy for identification of pigments”.
22. Lines 265-266, caption of Figure 10 – The sentence “The blue colorants used were identified as lapis lazuli, indigo, azurite, and mixtures of these colorants” is redundant. I suggest removing it.
23. Figure 10 – I assume that the caption of the figure is wrong, and moreover, very vague and confusing. It looks like the blue line on figure b) is a reference for lead white, not for indigo, or the key is wrong. I recommend checking it carefully. Moreover, the caption needs to be simplified, for example “Raman spectra of the blue colors (black lines) found in: a) LA152 (lapis lazuli reference - blue line), b) LA 193 and LA 253 (indigo reference - blue line), c) LA 216 (calcium carbonate reference - blue line), and d) details of the illuminations in LA 193 (left) and LA 216 (right); lapis lazuli (▼), indigo (♦), lead white (●) and calcium carbonate (▲)”, but at the same time, the symbols need be completed for each characteristic peak in the figures a, b and c. I suggest changing the caption of other figures in a similar way.
24. Figure 11 – I suggest adding orpiment and indigo markers on the figure a) and b), respectively.
25. Figure 12 – I suggest drawing the blue spectra of references above or below other three and label them d) and e). This will make the figure and its caption more clear. Moreover, peaks of vermilion do not have markers dedicated to vermilion.
26. Lines 298-299, caption of Figure 13 – The sentence “Orpiment and mixtures of this colorant with realgar and lead white were identified for the yellows” is redundant. I suggest removing it.
27. Line 315 – I suggest changing the heading into “Reds and pinks based on organic dyes”.
28. Figures 14, 15 and 16 – I suggest marking reference spectra with letters, for example c) and e) in Figure 14. It will need to reorganize the other markers in the figures as well.
29. Line 320 – The use of the word "identified" is a bit of an exaggeration. I propose to use “determined”, “distinguished” or “detected”.
30. Line 321 – It should be “see Figure C6, Appendix C”.
31. Appendix A – The reader can get lost in the abbreviations used: p., pp., f., ff., v, r - lack of uniformity in their use and explanation of their meanings.
32. Appendix C, Figure C1 – the caption is unclear. Maybe it would be a good idea to separate each subsequent spectrum description with semicolons.
33. Appendix C – The labels of the Y-axis in Figures C1-C3 are different, but the kinds of spectra and the way of its representation is the same, so it needs to be unified.
34. I suggest making all Appendix available directly on the publisher's website and not through dropbox, access to which may change or be lost.
In general, it can be easily seen that the manuscript was prepared/written by several people, because it lacks a coherent form, which is manifested, among others, in the in the presentation of figures and in various forms of results description (most comments above). However, definitely one more thing should be fixed, which is the inconsistency in the use of present and past tenses in different parts of Section 2.
Author Response
"The manuscript untitled “Exceptional illuminated manuscripts at the Gulbenkian Museum: a Bible and three Gospels produced in the Armenian diaspora” presents research on colors used in the illuminations of the manuscripts dated to the 17th century. By using fiber-optics reflectance spectroscopy and Raman spectroscopy Authors were able to identify pigments present in four objects produced by Armenian diaspora communities of Constantinople, Isfahan, and Crimea. I read the manuscript with an interest and I think it can be interesting also to those in non-destructive techniques in pigment analysis, human archaeology and conservation science. Nevertheless, I have comments and suggestions which I consider important and I kindly ask Authors to take it into account."
Answer: We thank the careful revision made by the reviewer and the constructive suggestions. Below is a point-by-point response; the text was revised accordingly. Moreover, we have rewritten certain sections, including the Painting technique and Conclusions, for clarity. The legends used in the spectra were also revised. The most relevant changes are marked in blue.
1) "The title does not introduce the type of research presented in the paper."
Answer: The title was corrected as follows:
Exceptional illuminated manuscripts at the Gulbenkian Museum: the colors of a Bible and three Gospels produced in the Armenian diaspora
2) "Authors presents thesis that organic reds were made of Armenian cochineal, but they did not present any specific proof of that."
Answer: The reviewer is absolutely correct. We have no evidence in this work that would allow us to determine the insect source. The dye detected is based on carminic acid, i.e, it is an anthraquinone-based dye of animal origin. Thus, in the text, we used one of these terms. The last sentence of the Introduction, section 1.1, was corrected as follows: "Our specific interest is in the organic-based colorants, with hues ranging from pink to carmine and purple, which in future work we will study in depth to verify whether they were made with Armenian cochineal [10]."
3) Section 2.1 – I suggest moving this section before the sections “Painting technique”, as well as changing the Appendix A into Appendix C.
Answer: Section 2.1. (The molecular palette used to produce the colors of the Bible and the Gospels) provides the main results obtained by integrating the data from sections 2.2 and 2.3. We understand it could follow section 2.3, but we think starting the "Results and discussion" with it is more appropriate.
4) Appendix B presents the areas of analysis by Portable Raman and by FORS. I suggest adding more detailed information about how many areas of each colour were examined on each page/object. Maybe Authors could put numbers or other code system of the areas in the presented pictures and give their detailed list below.
Answer: Thank you so much for your suggestion. We have added our methodology for the acquisition of data per color in the “Materials and Methods” section 3.1.
5) Appendix A – numbers of sections and subsections are confusing. I suggest numbering second-level headings as follows: 1.1, 1.2 etc., 2.1, 2.2 etc., 3.1, 3.2 etc., and 4.1, 4.2 etc.
Answer: This is what we have in Appendix A. For each manuscript we have:
- Bible LA 152
1.1. Painting technique
1.2. Degradation issues
6) Line 165 – I suggest removing the word "other" from this sentence.
Answer: This was done.
7) Line 172 – Authors indicated Appendix A, but in my opinion, it needs to be more specific, for example “(Appendix A, section 3.1)”
Answer: The reviewer is correct. We have added more precise information here and in other text parts.
8) Section 2.1 – whole section is a little difficult to follow. In my opinion, this section would benefit from presenting the similarities and differences in the pigments used in the manuscripts in a graphical form, e.g. in the form of a Venn diagram, radar plot or bar chart, using the actual number of areas studied or as a percentage (relative to the total number of areas in the object).
Answer: We do not have data on the areas of different colors for each manuscript at the moment. This would be very interesting work to carry out in the future, as we have already done it for studying Portuguese monastic production. For a researcher doing it for the first time, these four manuscripts will take at least a month to perform these calculations. Therefore, we prefer to keep this section as it is.
9) Lines 201-203 – the sentence “Lapis-lazuli and vergaut are the main blue and green colors in three of the manuscripts; in Gospel LA253 (17th century, Constantinople) azurite is the main blue and in Gospel LA193 malachite is the main green” in the caption of figure 6 is redundant. The same information is in section 2.1 (lines 189-192).
Answer: We found this information pertinent to make figure 6 self-sufficient in the text. So, we prefer to keep it.
10) Line 204 – why does "Gold" have its own unnumbered subsection in section 2.1 when other pigments/dyes do not.
Answer: Because gold is a metal that cannot be identified by Raman and FORS, but by using XRF. We added this information to it.
11) Lines 213-215, “Figure S1, Appendix C” – Authors refer to the same figure three times in only three lines of text. Is it necessary?
Answer: Now it is only referred to once.
12) In addition, the figures in Appendix C are numbered C1, C2, C3, etc., not S1, S2, S3. This need to be corrected throughout the whole manuscript.
Answer: This was corrected: now figures are numbered Sx.
13) Line 218, “(…) see Appendix B and C” and line 222, “Appendix C” – Please, add a precise reference to the figures.
Answer: This was done.
14) Line 222, “Appendix C” – Is this reference necessary?
Answer: Yes, the reference was completed, and it makes reference to two figures that present the two mixtures used to obtain a red and orange.
15) Lines 227-230 and Figure 7 – I suggest adding points a) and b) in the figure (as in Figure 11) and refer to them in the text. This will improve the clarity of this fragment of text. This comment also applies to the other figures in the main body of the manuscript and in Appendixes B and C (instead of using the words "left" and "right").
Answer: Thank you, this has been corrected.
16) Figure 7 – The caption is unclear and probably incorrect. What is the grey line on the left figure? The right figure shows only black and blue-lined reflectance spectra, no grey one.
Answer: Thank you, this has been corrected.
17) Figure 8 – I suggest changing part of the caption into “(…) pink color (black) and red color (grey), both from LA193 page 32v (…)”.
Answer: Thank you, this has been corrected.
18) Lines 236-239 – Were the obtained data compared with the references, as was the case in other cases? Why are the reference spectra not shown? Were they compared to the Armenian or Polish cochineal references, or even the American cochineal one? This is invariably important, especially in the context of the authors' thesis on the use of Armenian cochineal (see comment 2). Or maybe it would be worth discussing the differences between the spectra obtained for the studied areas and the references of lac dye, kermes or madder? Moreover, this part of the paragraph suggests that pinks and reds give differences in absorbance maxima, but the left part of Figure 8 contradicts this. Perhaps these differences in the spectra result from the use of different anthraquinone-based dye of animal origin or the degree of their degradation. Therefore, I think there are clear gaps at this part of the revised manuscript that need to be completed.
Answer: Legend for this figure, now figure 9, starts now as: Figure 9. Apparent absorbance spectra of animal anthraquinone-based paints.
We have also added a reference for American cochineal. The discussion of the insect source falls outside the scope of this work. We have made this clear in the introduction and throughout the text. It will be studied more in-depth in future work.
19) Figures 8, 9 and C6 (Appendix C) – the captions mention “cochineal” but in the main body of the manuscript it is called “anthraquinone-based dye”. Since it has not been conclusively proven that this was a specific type of cochineal, I suggest sticking with the latter one.
Answer: The reviewer is absolutely correct we have no evidence, in this work, which would allow us to determine the insect source. The dye detected is based on carminic acid, i.e, it is an anthraquinone-based dye of animal origin. Thus, in the text, we used one of these terms.
20) Lines 242-244 – The sentence “The areas of analysis where the best signals were obtained with FORS were selected and analysed with Raman spectroscopy to confirm and complete the information obtained” should be a new paragraph.
Answer: It is now a new paragraph.
21) Sections 2.2 and 2.3 – The headings could be changed and shortened. I suggest “Fiber-optics reflectance spectroscopy as a first screening for the study of pigments” and “Raman spectroscopy for identification of pigments”.
Answer: Headings were shortened as follows:
2.2. Fiber-optics reflectance spectroscopy as a first screening for the study of the colorants
2.3. Raman spectroscopy for identification of the colors of the Bible and the Gospels
22) Lines 265-266, caption of Figure 10 – The sentence “The blue colorants used were identified as lapis lazuli, indigo, azurite, and mixtures of these colorants” is redundant. I suggest removing it.
Answer: It was removed.
23) Figure 10 – I assume that the caption of the figure is wrong, and moreover, very vague and confusing. It looks like the blue line on figure b) is a reference for lead white, not for indigo, or the key is wrong. I recommend checking it carefully. Moreover, the caption needs to be simplified, for example “Raman spectra of the blue colors (black lines) found in: a) LA152 (lapis lazuli reference - blue line), b) LA 193 and LA 253 (indigo reference - blue line), c) LA 216 (calcium carbonate reference - blue line), and d) details of the illuminations in LA 193 (left) and LA 216 (right); lapis lazuli (▼), indigo (♦), lead white (●) and calcium carbonate (▲)”, but at the same time, the symbols need be completed for each characteristic peak in the figures a, b and c. I suggest changing the caption of other figures in a similar way.
Answer: Thank you for the observation. The caption has been improved as well as the captions for the following figures. The symbols have been applied accordingly to all the figures in question.
24) Figure 11 – I suggest adding orpiment and indigo markers on the figure a) and b), respectively.
Answer: Thank you, this has been corrected.
25) Figure 12 – I suggest drawing the blue spectra of references above or below other three and label them d) and e). This will make the figure and its caption more clear. Moreover, peaks of vermilion do not have markers dedicated to vermilion.
Answer: Thank you, this has been corrected.
26) Lines 298-299, caption of Figure 13 – The sentence “Orpiment and mixtures of this colorant with realgar and lead white were identified for the yellows” is redundant. I suggest removing it.
Answer: In the legend for figure 13, we found this information pertinent to make the figure self-sufficient in the text. So, we prefer to keep it.
27) Line 315 – I suggest changing the heading into “Reds and pinks based on organic dyes”.
Answer: It was done.
28) Figures 14, 15 and 16 – I suggest marking reference spectra with letters, for example c) and e) in Figure 14. It will need to reorganize the other markers in the figures as well.
Answer: Thank you, this has been corrected.
29) Line 320 – The use of the word "identified" is a bit of an exaggeration. I propose to use “determined”, “distinguished” or “detected”.
Answer: Thank you, this has been corrected.
30) Line 321 – It should be “see Figure C6, Appendix C”.
Answer: It was corrected in Appendix C, Figures S1 - S6.
31) Appendix A – The reader can get lost in the abbreviations used: p., pp., f., ff., v, r - lack of uniformity in their use and explanation of their meanings.
Answer: An explanatory note was added, at the beginning of Appendix A. It explains this irregularity of pages and folios, and gives their full names with respective abbreviations.
32) Appendix C, Figure C1 – the caption is unclear. Maybe it would be a good idea to separate each subsequent spectrum description with semicolons.
Answer: Thank you, the caption has been rewritten to be clearer.
33) Appendix C – The labels of the Y-axis in Figures C1-C3 are different, but the kinds of spectra and the way of its representation is the same, so it needs to be unified.
Answer: The label “Normalized reflectance” is used for a figure with two spectra because their values were normalized to enable the comparison of the bands, and the label “Reflectance” is used for the figures with just one spectrum because these last examples are not normalized. This information is needed so that the reader will know if the values of the spectra were subjected to any treatment.
34) I suggest making all Appendix available directly on the publisher's website and not through dropbox, access to which may change or be lost.
Answer: The reviewer is absolutely right. We will ask Heritage to assume the responsibility to keep all our Appendices on the publisher's website.
In general, it can be easily seen that the manuscript was prepared/written by several people, because it lacks a coherent form, which is manifested, among others, in the in the presentation of figures and in various forms of results description (most comments above). However, definitely one more thing should be fixed, which is the inconsistency in the use of present and past tenses in different parts of Section 2.
Answer: The manuscript has been completely revised to correct this inconsistency and to harmonize the different sections.

Reviewer 3 Report
The manuscript is very interesting. It just needs some minor revisions:
1. The term lapis lazuli. The blue pigment should be Ultramarine (commercial name) or Lazurite (blue mineral contained in the lapis lazuli). Lapis lazuli is a stone in which Lazurite is one of the minerals contained in this stone. The Raman spectrum in the manuscript appears to be the Raman spectrum of the Lazurite/ultramarine.
2. You use the term vermilion. Vermilion is the "commercial name" of the cinnabar. Do you have any information if the pigments used are natural? If natural, I suggest renaming vermilion as cinnabar.
3. the band at ca. 1087 cm-1 can be assigned to carbonate mineral species. To identify the calcium carbonates, you need to indicate the secondary bands. As in the literature, Ca, Mg, and Fe - carbonates have the main band at around 1087 cm-1. So, I suggest reviewing the plots and adding the due references.
Author Response
The manuscript is very interesting. It just needs some minor revisions.
Answer: We thank the helpful suggestions. Below is a point-by-point response; the text was revised accordingly. Moreover, we have rewritten certain sections, including the Painting technique and Conclusions, for clarity. The most relevant changes are marked in blue
1) The term lapis lazuli. The blue pigment should be Ultramarine (commercial name) or Lazurite (blue mineral contained in the lapis lazuli). Lapis lazuli is a stone in which Lazurite is one of the minerals contained in this stone. The Raman spectrum in the manuscript appears to be the Raman spectrum of the Lazurite/ultramarine. ”.
Answer: Thank you, this has been corrected.
In the context of medieval manuscript illuminations, the term lapis lazuli is used; possibly, the purest stones were ground and applied as paints. They, too, could be purified. In the Raman spectra, the characteristic sulfur clusters of lazurite are identified, but for the sake of simplicity, we prefer to use references to compare with the acquired spectra. Therefore, in this context, we prefer to keep the term lapis lazuli and add the information that lazurite is detected by Raman spectroscopy.
2) You use the term vermilion. Vermilion is the "commercial name" of the cinnabar. Do you have any information if the pigments used are natural? If natural, I suggest renaming vermilion as cinnabar. ”.
Answer: In the context of medieval manuscript illuminations, the term vermilion is used when we do not know whether it was produced using cinnabar stone or by synthesis.
3) the band at ca. 1087 cm-1 can be assigned to carbonate mineral species. To identify the calcium carbonates, you need to indicate the secondary bands. As in the literature, Ca, Mg, and Fe - carbonates have the main band at around 1087 cm-1. So, I suggest reviewing the plots and adding the due references. ”.
Answer: Thank you, this has been corrected.
